# Biomarkers in the Diagnosis of Endometrial Precancers. Molecular Characteristics, Candidate Immunohistochemical Markers, and Promising Results of Three-Marker Panel: Current Status and Future Directions

**DOI:** 10.3390/cancers16061159

**Published:** 2024-03-15

**Authors:** Shuang Niu, Kyle Molberg, Diego H. Castrillon, Elena Lucas, Hao Chen

**Affiliations:** 1Department of Pathology, UT Southwestern Medical Center, Dallas, TX 75390, USA; shuang.niu@utsouthwestern.edu (S.N.); kyle.molberg@utsouthwestern.edu (K.M.); diego.castrillon@utsouthwestern.edu (D.H.C.); 2Department of Pathology, Parkland Health, Dallas, TX 75235, USA; 3Department of Obstetrics and Gynecology, UT Southwestern Medical Center, Dallas, TX 75390, USA; 4Harold C. Simmons Comprehensive Cancer Center, UT Southwestern Medical Center, Dallas, TX 75390, USA

**Keywords:** endometrial carcinoma, precancer, AH/EIN, biomarkers, accurate diagnosis

## Abstract

**Simple Summary:**

Endometrial carcinoma is a common type of cancer in women, ranking as the most frequent gynecological cancer and the fourth most common overall. Its occurrence has been rising steadily, becoming a serious health concern. Detecting its immediate precursors, known as precancers, is crucial for saving lives. In recent years, research into the genetic makeup of endometrial carcinoma and its precancers has progressed, especially focusing on the most common subtype called endometrioid. New biomarkers have been discovered that could help in identifying precancers, potentially improving diagnosis and treatment. This review summarizes these recent discoveries and their significance in diagnosing precancers of endometrial carcinoma.

**Abstract:**

Endometrial carcinoma stands as the most prevalent gynecological cancer and the fourth most common cancer affecting women. The incidence of endometrial cancer has been steadily increasing over the past decade, posing a significant threat to public health. The early detection of its precancers remains a critical and evolving concern to reduce mortality associated with endometrial carcinoma. In the last decade, our understanding of endometrial carcinoma and its precancers has advanced through systematic investigations into the molecular genetics of endometrial carcinoma and its precancers. In this review, we focus on advances in precancers associated with the endometrioid subtype, by far the most common histologic variant of endometrial adenocarcinoma. Recent investigations have led to the identification of new biomarkers, and the proposed incorporation of these biomarkers or biomarker panels into the diagnostic framework of endometrial carcinoma precancers. Here, we review these recent advances and their relevance to the histopathologic diagnosis of endometrial carcinoma precancers.

## 1. Introduction

Endometrial carcinoma, the most common gynecological malignancy in the United States [1,2], remains one of the biggest threats to women’s health despite advances in treatment options [3]. Over the last decade, large-scale genomic analyses of endometrial carcinoma have significantly advanced our understanding of the disease, leading to a new molecular classification of endometrial carcinoma that improves risk stratification and patient management. The early and accurate detection of its precancers—precursor lesions with a high risk for progression to cancers—is of paramount importance, offering a valuable opportunity for a cure before progression to full-fledged cancer. This, in turn, could reduce the prevalence of endometrial carcinoma and lower healthcare costs associated with its treatment. However, accurately diagnosing endometrial carcinoma precancers remains one of the most challenging aspects of gynecological pathology.

Genomic studies have significantly enhanced our knowledge of the underlying molecular alterations of endometrial carcinoma precancers. These studies have not only deepened our understanding of precancerous genetic changes but have also unveiled promising biomarkers based on these genetic aberrations that can aid in improving diagnostic accuracy of these lesions. Investigations into biomarkers and biomarker panels have yielded promising results, leading to the first introduction of biomarkers into the diagnostic schema in the updated 2020 World Health Organization (WHO) Classification of Female Genital Tumors [4]. This review covers several key aspects of endometrial precancers. It explores the molecular genetics of these lesions along with their associations with respective molecular endometrial cancer groups, examines candidate biomarkers for precancer, and focuses on the utility of a three-marker panel in the accurate diagnosis of atypical hyperplasia/endometrioid intraepithelial neoplasia (AH/EIN).

## 2. Molecular Classification of Endometrial Carcinoma

One of the most significant advances in gynecological cancers, the molecular classification of endometrial carcinoma, has demonstrated substantial prognostic value, greatly impacting patient risk stratification and management [5,6,7,8]. Initially proposed by The Cancer Genome Atlas (TCGA) based on mutation profiles, microsatellite stability, somatic copy number alterations, and tumor mutation burden, endometrial carcinoma was classified into the following four molecular groups: (1) DNA polymerase ɛ (*POLE*)-mutant (*POLE*mut/ultramutated); (2) microsatellite instability (MSI)-high (hypermutated); (3) copy number low; and (4) copy number high/serous-like carcinomas. Tumors in the first group (*POLE*mut/ultramutated) have an excellent prognosis regardless of the tumor grade and stage. In contrast, tumors in the fourth group (copy number high/serous-like) show nearly universal *TP53* mutations and the worst prognosis among the four groups. The second and third groups (MSI-high/hypermutated and copy number low) are associated with an intermediate prognosis.

While not perfect, a significant association exists between molecular and histological classifications of endometrial carcinomas. Molecular groups (1–3) roughly align with endometrial endometrioid adenocarcinoma, while the copy number high (serous-like) group is significantly associated with endometrial serous and mixed histology carcinomas. Approximately 25% of FIGO grade 3 endometrioid carcinomas are also classified as copy number high tumors (molecular group 4). As illustrated in Figure 1, the first three molecular groups had few *TP53* mutations but frequent mutations in *PTEN*, *CTNNB1*, *ARID1A*, *PIK3R1*, *PIK3CA*, and *KRAS*. Mutations in these genes lead to disruption of intracellular signaling pathways critical for endometrial cell growth (including PI3K–PTEN–AKT–mTOR, RAS–MEK–ERK, and canonical WNT–ß-catenin) and the SWI/SNF chromatin remodeling complex. In contrast, the copy number high (serous-like) group had frequent *TP53* mutations, with numerous chromosomal rearrangements and chromosome-level instability but fewer single nucleotide variants, representing point mutations in these cancer drivers commonly found in the other three molecular subgroups [8].

Despite the efficacy of the TCGA classification, high cost and lack of availability of next generation sequencing (NGS) assays in many clinical settings has limited its widespread implementation. To address such challenges, several studies [7,9,10] explored alternative more practical classification strategies using surrogate immunohistochemical (IHC) markers (e.g., p53, MSH6, and PMS2) or targeted assays for *POLE* mutation based on Sanger sequencing or targeted NGS (*POLE* exons 9–14 or 9, 13, and 14). This alternative approach categorized endometrial carcinoma into the following four groups: *POLE*mut; mismatch repair (MMR) deficient (MMRd); no specific molecular profile (NSMP); and abnormal p53 (p53abn). Multiple studies [5,7,9,10,11] have validated the prognostic value of this alternative approach. Nonetheless, the reliable identification of cancer-driving *POLE* mutations in a cost-effective manner remains a significant challenge, although several *POLE* assays not based on NGS have been described that should be capable of reliably identifying *POLE* mutations [12]. It is important to note that, while the alternative classification bears similarity to the original TCGA classification, they are not identical. Additionally, instances of tumors with characteristics of multiple classifiers—such as POLE mutations, MMR deficiency, and abnormal p53—have also been observed. Consequently, the final classification hierarchy of multiple classifiers prioritizes *POLE*mut > MMRd > p53abn [13].

## 3. Diagnostic System of the Precancers of Endometrial Adenocarcinoma

In the WHO 2020 classification, the only recognized morphological precancerous lesion is atypical hyperplasia/endometrioid intraepithelial neoplasia (AH/EIN). It is considered the precancer for endometrioid carcinoma, as indicated by its name. The early literature used the term “endometrial intraepithelial neoplasia” [14], later modified to “endometrioid intraepithelial neoplasia”, to emphasize the distinct endometrioid pathway of carcinogenesis [15,16]. The morphological diagnostic criteria in the WHO 2020 classification include (1) glandular crowding (gland/stroma ratio > 1), (2) cytology (nuclear morphology with or without cytoplasmic features) differs between lesion and the background endometrium, (3) reasonable size of the lesion to exclude artifactual crowding, and (4) exclusion of carcinoma or mimics such as endometrial polyps. One of the most significant developments in the WHO 2020 characterization of endometrial cancer precursors is the introduction of biomarkers as diagnostic tools for AH/EIN. The loss of immunoreactivity for PTEN, PAX2, or MMR is specified as a “desirable” criterion for diagnosis [4]. However, the inclusion of these markers vs. the exclusion of other potentially useful markers is not clearly rationalized.

Serous endometrial intraepithelial carcinoma (SEIC), once considered the precancer to endometrial serous carcinoma, is now recognized as an early form of endometrial serous carcinoma rather than a precancer, as it can be associated with extra-uterine metastatic spread even in the absence of demonstrable stromal invasion [17,18]. A sequential genetic/morphological events model (i.e., from endometrial p53 signature lesion to endometrial glandular dysplasia (EmGD) to SEIC to ESC) similar to that of tubo-ovarian high-grade serous carcinoma has been proposed for the development of ESC [19,20]. Considering the well-established association between SEIC and ESC, the proposed model accounts for the majority of p53abn endometrial carcinomas.

## 4. Genetic Characteristics of AH/EIN and Its Association with Carcinoma 

Recent studies investigating genome-wide mutational profiles in paired AH/EIN and carcinoma have shed light on their genetic association between them, particularly in the context of molecular classification. In AH/EIN, the most commonly mutated genes, including *PTEN*, *CTNNB1*, *ARID1A*, *PIK3CA*, and *KRAS*, are also mutated in endometrial carcinoma [21,22,23,24,25]. While the majority of AH/EIN cases fall into the NSMP group, a small percentage would be categorized within the *POLE*mut, MMRd, and p53abn groups [21,24,25]. Immunohistochemistry (IHC) studies on AH/EIN further confirm that a minority of cases exhibit defective MMR and mutant-type p53 expression [24,25,26,27]. Thus, AH/EIN likely serves as the precancer of endometrial carcinoma within the first three molecular groups and at least a fraction of the p53abn group (as depicted in Figure 2). Interestingly, while a stepwise acquisition of cancer driver mutations and progressive accumulation of tumor mutational burden is observed in the majority of cases transitioning from AH/EIN to endometrial carcinoma, private mutations (present in only one sample) are not uncommon [21,24]. Therefore, while the linear pathway involving the stepwise accumulation of the molecular events accounts for the endometrial carcinoma carcinogenesis in the majority of cases, a more complex pathway may exist, wherein AH/EIN and carcinoma diverge early and develop independently [21,28]. Intra-tumor heterogeneity and sampling issues also likely account for variations among samples from the same patient. A rare case involving two distinct AH/EIN clones within a single endometrial biopsy has been reported, supporting the presence of parallel pathways [29]. 

## 5. Candidate Immunohistochemical Biomarkers for Diagnosing AH/EIN

In the past decade, the search for biomarkers that may aid in the accurate diagnosis of AH/EIN has been an intense area of research in gynecological pathology. Advancement of molecular studies on endometrial carcinoma and its precancers have led to the exploration of numerous IHC biomarkers, proposed as diagnostic adjuncts to H&E staining, demonstrating their practical utility. Notably, many of those candidate markers were derived from the TCGA endometrial carcinoma study (highlighted in Figure 1). In the following sections, we offer a concise review of biomarkers alongside their performances in AH/EIN. Continuous efforts have been directed on refining the interpretation criteria for individual biomarkers by comparing the expression patterns between normal endometrium and AH/EIN. Detailed information regarding the expression pattern of individual biomarkers in normal endometrium, as well as the interpretation criteria for aberrant expression in AH/EIN, is presented in Table 1.

### 5.1. PAX2

*PAX2*, a member of the large paired box gene family, has been utilized as a marker of Müllerian duct derivatives [30]. As a DNA-binding transcription factor, PAX2 is exclusively localized within the nuclei. The aberrant loss of nuclear expression of PAX2 occurs in approximately 70–80% of endometrial carcinoma and AH/EIN [31]. The initial report documenting frequent immunohistochemical loss of PAX2 in AH/EIN dates back to 2010, as reported by G. Mutter’s group [31]. Additionally, PAX2 loss is a very early (if not initiating) event in the development of AH/EIN [8,24,31,32,33,34,35,36,37,38,39,40]. The mechanism of this loss is poorly understood, as *PAX2* is not one of the frequently mutated genes identified in endometrial carcinoma [8]. It has been suggested that this may be due to epigenetic modifications of *PAX2* [41], similar to the process of *PAX2* gene suppression that occurs in the later stages of embryonic development [42]. Several investigations have confirmed the utility of PAX2 loss in the diagnosis of AH/EIN [15,31,43,44,45,46]. It is noteworthy that sporadic PAX2 loss in scattered glands is not uncommon in benign endometrium. However, the loss is typically focal and does not exceed 5% of the entire specimen [26,31,33], except in cases of endometrial polyps, where approximately 10–20% of cases show more widespread PAX2 loss [47,48]. When results of PAX2 IHC are applied appropriately in the context of morphologic findings, the substantial loss of nuclear expression of PAX2 in AH/EIN is highly specific (95%) and demonstrates good sensitivity (72%), as indicated in a recently published meta-analysis [49]. More importantly, the expression pattern of PAX2 appears to be persistent in progestin-treated residual AH/EIN [50], making it a potential marker for identifying residual disease in endometrial biopsies after progestin treatment.

### 5.2. PTEN

PTEN, a ubiquitously expressed lipid phosphatase, is present throughout the cell in the nucleus, cytoplasm, and cell membrane, and is broadly expressed across cell types including endometrial epithelia, stroma, and leukocytes [51]. As the most frequently mutated gene in endometrial carcinoma and AH/EIN [8,35,52,53], PTEN is a tumor suppressor negatively regulating the PI3K/AKT/mTOR pathway [54]. *PTEN* point mutations are believed to represent early driver events, as they frequently occur in AH/EIN [8,55]. Initial studies documenting PTEN loss in AH/EIN were conducted by Mutter’s group [31,56,57]. Like PAX2, sporadic PTEN loss in scattered glands is also common in benign endometrium. However, such loss is typically focal and does not exceed 5% of the entire specimen [26,47,48]. The intensity of PTEN expression in normal endometrial glands varies from strong to weak. Substantial loss of PTEN expression occurs in approximately 50% of AH/EIN [26,31,33,47,48] (it is important to not score weak expression as loss). While the frequent loss of PTEN expression aligns with the high frequency of *PTEN* point mutations in endometrial carcinoma, loss is probably not entirely explained by point mutations. Intragenic deletions appear to be common within the *PTEN* locus, and epigenetic modifications such as promoter hypermethylation may result in epigenetic silencing of *PTEN* [31,56,58,59]. Similar to PAX2, the substantial loss of PTEN expression persists following progestin treatment [43], potentially aiding in the identification of residual disease.

### 5.3. β-Catenin

Beta-catenin (β-catenin) is a key constituent of the canonical WNT–β-catenin pathway, which regulates a variety of cellular processes, including cell and tissue proliferation, differentiation, and carcinogenesis [60,61]. In cancers, *CTNNB1* (encoding the β-catenin protein) may develop gain-of-function mutations, altering specific residues within exon 3. These residues, constituting a β-catenin protein degradation motif, prevent phosphorylation and subsequent ubiquitin-mediated proteasomal degradation of β-catenin. This alteration results in protein overexpression and its abnormal relocalization from the membrane/cytoplasm to the nucleus [62,63,64]. *CTNNB1* is one of the most frequently mutated genes in endometrial carcinoma, and mutations occur at an unusually high frequency (~50%) in microsatellite stable, copy number low, and endometrioid carcinomas [8]. In the normal endometrium, β-catenin exhibits ubiquitous cell membrane expression [26,46,47,48], with the exception of some cases of the interval phase endometrium, where weak nuclear localization may be observed [26,29]. In contrast, strong nuclear β-catenin localization, typically associated with overall overexpression, serves as a reliable indicator of β-catenin activation in AH/EIN or endometrial carcinoma, demonstrating ≥90% specificity and sensitivity for *CTNNB1* mutations [63,64,65]. Notably, the nuclear staining can be patchy and focal, with nearly half of *CTNNB1* mutant cases showing nuclear staining in only 5–10% of tumor cells [63,64]. Several small studies on β-catenin expression in AH/EIN showed that 26–50% of AH/EIN cases demonstrated a β-catenin nuclear-staining pattern [24,26,46,47,66,67,68]. *CTNNB1* point mutations are early events in endometrial carcinoma development, making β-catenin (along with PAX2 and PTEN) a plausible biomarker for endometrioid precancers [34,69]. The 2017 study by Kurnit et al. demonstrated significant associations between *CTNNB1* mutations and an increased risk of recurrence among low-grade early-stage endometrial carcinomas [66]. Thus, the presence of *CTNNB1* mutations in endometrial carcinoma appears to be a negative prognostic factor.

### 5.4. Mismatch Repair (MMR) Proteins

Mismatch repair (MMR) protein deficiency (MMRd) is detected in approximately 20–30% of all endometrial carcinomas and is primarily associated with endometrioid histology [70,71]. Combined deficiencies in MLH1 and PMS2 account for approximately half of MMRd endometrial carcinomas, the majority of which are caused by promoter hypermethylation of the *MLH1* gene, with about 10% of cases caused by somatically acquired mutations [72,73]. Other MMR protein deficiencies (PMS2 only, dual MSH2/MSH6, or MSH6 only) account for the remainder of MMRd endometrial carcinomas, with most cases originating from patients with Lynch syndrome [27]. Pathologists can reliably identify MMR protein deficiency in AH/EIN using the standard scoring system in Lynch syndrome screening for newly diagnosed endometrial cancers [73,74,75,76]. Studies investigating MMR expression in unselected biopsies with AH/EIN report that less than 5% of AH/EIN cases demonstrate loss of MMR expression, a surprisingly low number given the high incidence of MMRd in invasive cancers [24,26,77,78]. While highly specifically differentiating AH/EIN from non-neoplastic mimics, the utility of MMR in diagnosing AH/EIN remains debatable due to the low prevalence of MMR deficiency in AH/EIN. One plausible explanation is that MMRd precursors might rapidly progress to endometrial carcinomas, making their clinical detection less likely [79]. Another potential explanation involves MMR deficiency due to MLH1 methylation, which could be a later event in the evolution of endometrial carcinoma. 

### 5.5. ARID1A

*ARID1A* (also known as BAF250A), an important component of the SWI/SNF nucleosome remodeling complex, is also one of the most frequently mutated genes in endometrioid carcinoma (illustrated in Figure 1) [80,81,82]. ARID1A is ubiquitously expressed in the normal endometrium. *ARID1A* mutations often lead to the complete loss of protein expression [83]. Approximately 10% of AH/EIN demonstrated clonal loss of ARID1A expression [82,83]. Studies showed that the clonal loss ARID1A expression is highly associated with ARID1A mutations [24,84]. A recently published meta-analysis of ARID1A in AH/EIN also showed that ARID1A loss is highly specific as a diagnostic marker for AH/EIN [85].

### 5.6. p53

p53 aberrancy is the defining feature of the p53abn group in the molecular classification of endometrial carcinoma, which roughly correlates with endometrial serous carcinoma. However, it is well-known that *TP53* is also mutated in some endometrioid adenocarcinomas, commonly associated with POLEmut and/or MMRd tumors [13]. In the presence of *POLE* pathogenic mutation or MMR deficiency, p53 alterations are considered as secondary events acquired during tumor progression [13,69]. Since subclonal *TP53* alterations are likely acquired late during endometrioid carcinoma progression, the likelihood of p53 aberrancy in AH/EIN, an early stage of endometrial carcinoma development, is likely low, as further confirmed by studies [26,86]. Pathologists are generally familiar with the identification of p53 mutation patterns [87], which include (1) overexpression, characterized by diffuse, strong nuclear positivity involving at least 80% of tumor cells; (2) “null” pattern, characterized by complete absence of nuclear staining; (3) cytoplasmic, characterized by cytoplasmic staining accompanied by variable nuclear staining; and (4) heterogeneous pattern, in which different clones show different staining patterns [87]. 

## 6. Multi-Marker Panels for Identifying AH/EIN

As no single candidate biomarker proved capable of detecting all or nearly all cases of AH/EIN, substantial efforts have concentrated on enhancing detection sensitivity through the use of a multi-marker panel. A recent study by Aguilar et al. evaluated the efficacy of a three-marker panel in identifying AH/EIN. The rate of individual marker positivity in AH/EIN varied with PAX2 exhibiting better sensitivity (81%) than PTEN (51%) or β-catenin (48%). Notably, the majority of cases (70%) exhibited aberrant expression in at least two markers, enhancing diagnostic reliability. When applied collectively, the combination of PAX2, PTEN, and β-catenin achieved a significantly improved sensitivity of 92.8% in bona fide AH/EIN detection. Employing a refined interpretation method with the nuanced knowledge of these markers’ expression patterns in normal endometrium (outlined in detail in Table 1) is needed to attain high specificity rates [26]. Consequently, this panel proved to serve as a valuable diagnostic adjunct in histopathologic diagnoses of AH/EIN.

### 6.1. Morules as Surrogate Markers for β-Catenin Mutation

Morules, recognized as distinct morphological features often termed “squamous morules” or “morular metaplasia”, manifest as variably sized solid nests comprising spindled or ovoid cells with indistinct cell borders. Frequently observed in association with malignancies or precancerous conditions [87,88,89,90], the presence of morules is strongly associated with *CTNNB1* mutations. Accordingly, they ubiquitously demonstrate aberrant β-catenin expression. Conversely, more than 50% of AH/EIN cases exhibiting β-catenin aberrancy contain morules. These findings underscore morules as reliable indicators of *CTNNB1* mutations in both endometrial adenocarcinoma and AH/EIN [67], establishing their significance as a morphological “marker” for AH/EIN in cases with questionable or subtle findings. Given their easily identifiable nature, morules serve as a valuable diagnostic clue for AH/EIN, especially in scenarios where β-catenin testing might not be readily accessible in routine practice. An AH/EIN case with morules and aberrant expression of PAX2, PTEN, and β-catenin is presented in Figure 3.

### 6.2. Utility of Three-Marker Panel Plus Morules in Diagnosis of Challenging AH/EIN

The diagnosis of AH/EIN presents challenges in various scenarios, including its presence within endometrial polyps (referred to as AH/EIN in polyps), AH/EIN with secretory changes (secretory AH/EIN), small-sized AH/EIN (small lesions of uncertain significance/focal gland crowding), and cases of AH/EIN subjected to progestin treatment (illustrated in Figure 4). 

#### 6.2.1. Utility of Three-Marker Panel Plus Morules in AH/EIN in Polyps 

Within polyps, differentiating between benign glandular crowding and metaplastic changes typical in endometrial polyps and AH/EIN poses a diagnostic challenge. Pathologists often grapple with the “mimics exclusion” criterion’s application as glands are abnormal and can exhibit increased crowding and there are no clear-cut criteria for allowable gland crowding in polyps. A recent study by Lucas et al. demonstrated that the prevalence of PAX2 and PTEN aberrancy was comparable between AH/EIN in polyps and non-polyp AH/EIN (64.8% vs. 81.1% and 39.0% vs. 50.5%, respectively), whereas β-catenin aberrancy and the presence of morules was significantly higher in AH/EIN in polyps than in non-polyp AH/EIN (61.9% vs. 47.7% for β-catenin and 38.1% vs. 24.3% for morules). The sensitivity of the combined three-marker panel for detection of AH/EIN in polyps was similar to that in non-polyp AH/EIN (92.4% vs. 92.8%) [47]. Care should be exercised when interpreting individual markers. While benign polyps typically demonstrate normal expression of PTEN and β-catenin, aberrant PAX2 expression can be observed in up to 15% of polyps [48]. Similarly, morules can also be rarely encountered in benign polyps. Thus, the diagnosis of AH/EIN in polyps should be made with caution and only in the appropriate morphological context. 

#### 6.2.2. Utility of Three-Marker Panel Plus Morules in Secretory AH/EIN 

Secretory AH/EIN poses difficulties due to the challenges in applying glandular crowding criterion, as this is also the feature of secretory phase endometrium, and the less pronounced or absent nuclear and cytological atypia under the influence of progesterone [88]. Our study on a large cohort of benign secretory endometrium and secretory AH/EIN demonstrates the utility of a three-marker panel and morules in the diagnosis of AH/EIN, with sensitivity comparable to that in bona fide AH/EIN (unpublished data). An interesting finding is that diffuse or focal weak nuclear localization of β-catenin can be seen in the majority of interval-phase endometrium cases, confirming our previous observations [26,29]. The exact mechanism of this weak nuclear localization is still unknown, although it is not associated with mutations. Nonetheless, by applying the “strong distinct nuclear expression (same intensity as intervening cell membranes)” criterion, it is possible to reliably distinguish true β-catenin aberrancy from the weak nuclear expression in interval-phase endometrium. 

#### 6.2.3. Utility of Three-Marker Panel in Small-Sized AH/EIN

The characterization of small-sized AH/EIN is challenging, primarily due to variations in glandular architecture, density, and cytology observed during normal menstrual cycling, making precise size criterion assessment complex. The term “small-sized AH/EIN” refers to lesions typically measuring nearly or less than 1 mm. Originally, the criterion “≥ 1 mm”, derived from the morphometric study [89] aimed at excluding artifactual crowding, was modified in the WHO 2020 system to a “reasonable size of the lesion to exclude artifactual crowding”. Consequently, many previously subdiagnostic lesions, often referred to as “focal glandular crowding”, could be categorized as AH/EIN per the WHO 2020 system. Obviously, the interpretation criteria for PAX2 and PTEN, requiring “the absence of staining in > 5% of glands”, do not directly apply to these lesions, given their representation <5% of the entire specimen in most cases. Thus, to assess the aberrancy of these markers, it is appropriate to assess them only in the crowded focus in question, where >50% loss of expression is considered aberrant. In our practice, we have found that appropriately applying this criterion, combining marker aberrancy with morphological “clonal” demarcation from the background effectively differentiates between clonal alteration and sporadic alterations in isolated glands (illustrated in Figure 5A–C). While a comprehensive large-scale study validating the utility of the three-marker panel in small-sized AH/EIN is yet to be conducted, our preliminary study, though limited in cases, suggests its effectiveness in this context [78,90].

#### 6.2.4. Utility of Three-Marker Panel Plus Morules in the Identification of Residual Lesion in Progestin-Treated AH/EIN

Progestin-treated AH/EIN adds another layer of difficulty, as progestin treatment induces notable morphological changes in both neoplastic and non-neoplastic endometrium. This can mask classic diagnostic features of AH/EIN, including glandular architecture and nuclear atypia, even when persistent or residual disease is present [88]. A study by Wheeler et al. suggested that persistent architectural abnormalities and/or nuclear atypia had predictive value for treatment failure (residual EIN/AH) [91]. However, in practical scenarios post-treatment, nuclear atypia is frequently absent or markedly reduced, whereas architectural complexity commonly persists in post-treatment endometrial samples. A longitudinal study by Chen et al. on consecutive endometrial biopsies from a large cohort of patients diagnosed with AH/EIN on continuous progestin treatment demonstrated the persistent aberrancy pattern of PAX2 and PTEN between pre- and post-treatment lesions [50]. As β-catenin aberrancy is caused by mutations, it is reasonable to believe that β-catenin aberrancy also persists in post-treatment residual lesions, which was confirmed in a small number of cases [24,68]. The presence of morules is particularly useful in progestin-treated AH/EIN. Unlike the glandular component in AH/EIN and endometroid carcinoma, morules tend to be resistant to progestin treatment and persist or even markedly increase after treatment [92,93]. It has been postulated that this may be due to a lack of progesterone receptors in these structures.

Performing a three-marker panel on all initial endometrial biopsies or curettages with the diagnosis of AH/EIN may be beneficial for establishing a baseline immunoprofile of the lesion, which can be useful with conservative (i.e., medical) management. Information regarding the aberrancy of any of the three markers may aid in identifying residual lesions after progestin treatment. Residual lesions, especially those in diagnostically challenging cases, tend to be small. The same criterion for small-sized AH/EIN, which requires a significant overlap between marker aberrancy and suspected residual lesion (e.g., areas with architectural complexity and/or nuclear atypia), proves to be useful in such situations (as illustrated in Figure 5D–F).

#### 6.2.5. Utility of Three-Marker Panel in Other Putative Endometrial Precancers 

Atypical polypoid adenomyoma (APA) and papillary mucinous proliferations (PMP) are uncommon polypoid endometrial lesions. APA is characterized by atypical architecturally complex endometrial glands within benign myomatous or fibromyomatous stroma, often accompanied by morules [94,95]. Despite its initial classification as a benign lesion, subsequent research has revealed an association with an elevated risk of progressing to endometrial adenocarcinoma [96,97,98]. PMP consists of irregular or complex mucinous glands displaying distinctive intraluminal papillary tufts and the absence of nuclear atypia. PMP is recognized as a potential precancerous lesion for endometrial carcinoma with mucinous differentiation as it frequently harbors KRAS mutations [99,100,101,102]. A significant proportion of APAs and PMPs exhibited one or more marker aberrancies [47]. This finding suggests that APA and PMP may represent distinctive morphologic variants within the AH/EIN spectrum.

## 7. Limitations of Three-Marker Panel

In light of the demonstrated or potential benefits of the three-marker panel, it is essential to acknowledge several limitations. These constraints underscore the necessity for future studies to address and overcome these challenges. 

### 7.1. Detection Ceiling of Marker Panel 

While the three-marker panel demonstrates commendable performance (achieving over 90% accuracy in identifying AH/EIN), approximately 10% of cases exhibit no marker aberrancy. Inclusion of other known candidate markers, including ARID1A, MMR, or p53, into the panel did not increase the sensitivity beyond that of the three-marker panel, as cases with aberrancy in these additional markers had already been identified by PAX2, PTEN, or β-catenin, indicating a detection ceiling for the marker panel had been reached [26]. To increase sensitivity, the discovery of new markers for inclusion into the existing panel or alternative diagnostic approaches may be needed.

### 7.2. Resolution Ceiling of PAX2 and PTEN

Given the frequent occurrence of sporadic loss of expression for PAX2 and PTEN in normal endometrium, the criteria for considering aberrancy focuses on complete loss in more than 5% of glands. Although this criterion proves effective in larger or diffuse lesions, its utility diminishes when dealing with small-sized lesions. Notably, progestin-treated endometria frequently contain residual lesions comprising less than 5% of the tissue. The application of the “loss of expression in 50% of the lesion in question” criterion, may be helpful in certain instances. However, we have also encountered challenging cases, such as those where an exact overlap is not present or there is loss of expression in only a portion of the lesion of interest. In our anecdotal observations, we have noted that confidence in making an accurate diagnosis tends to decrease as the size of the lesion decreases.

### 7.3. Potential Advantages of an Expanded Panel 

While the inclusion of these markers does not necessarily improve overall sensitivity, expanding the existing three-marker panel with MMR (PMS2 and MSH6) and p53 may offer several potential benefits of enhanced specificity. The aberrancy of ARID1A, MMR (PMS2 and MSH6), and p53 is highly specific for a neoplastic process and is easily interpretable by most pathologists. The addition of these specific markers to the panel increases the likelihood of multi-marker aberrancy for each case. This, in turn, enhances confidence in the diagnosis, especially crucial for small-sized lesions where the interpretation of PAX2 or PTEN may be uncertain. Furthermore, the expanded panel opens the door to the potential for further classification of AH/EIN based on IHC/molecular signatures, aligning with the molecular classification of endometrial carcinoma. Given the prognostic value demonstrated by the molecular classification of endometrial carcinoma, a similar approach for AH/EIN could have significant clinical implications. This includes assessing the risk of progression to cancer and resistance to progestin treatment, ultimately impacting current clinical practices. In addition, incorporating MMR into the multi-marker panel may be beneficial for the early detection of patients with Lynch syndrome. This proactive approach may have broader implications for patient care, enabling timely intervention and management strategies. The additional markers (ARID1A, MMR, and p53) can be ordered together with three main markers or a reflex test.

## 8. Unresolved Issues and Future Directions 

### 8.1. Uncertain Significance of β-Catenin Aberrancy and Isolated Morules in Seemingly Normal Endometrial Glands during or after the Cessation of Progestin Therapy

Aberrant β-catenin nuclear staining can be present in biopsies with scattered seemingly “normal” glands in endometria previously diagnosed as AH/EIN during or after the cessation high-dose progestational agents (Figure 6A–F). Considering the direct association between β-catenin aberrancy and CTNNB1 mutations—a well-known tumor driver—it is reasonable to assume the presence of mutant CTNNB1 in these glands. However, it remains unclear whether these glands might progress to eventual tumor formation, challenging the appropriateness of categorizing them as residual or recurrent lesions. Similarly, isolated morules—defined as morules without associated glands—can occasionally be seen as the sole abnormal finding in biopsies during or after the cessation of progestin treatment (Figure 7A,B). Studies have supported a common lineage for neoplastic glands and morules showing identical mutational profiles [68,93]. This supports the notion that morules likely represent a neoplastic phenomenon within host AH/EIN. Of note, isolated morules can occasionally be observed in “benign” endometrium without progestin influence, which carries an increased risk of progressing into endometrial carcinoma [93]. However, the question whether morules represent residual lesions capable of progressing to bona fide AH/EIN remains unresolved. Long-term follow-up of a large patient cohort is necessary to determine their clinical significance.

### 8.2. Discordance between Morphology and Marker Aberrancy

As mentioned above, by appropriately applying the interpretation criteria and aberrancy thresholds, no abnormal marker expression is typically present in normal endometrium (whether proliferative or early-, mid-, or late-secretory), or in endometrial polyps, with the exception of PAX2, which can be aberrant in 10–20% of endometrial polyps [43,44]. However, marker aberrancy can be present in subdiagnostic lesions that fall short of meeting the criteria for AH/EIN, disordered proliferative endometria, or nonatypical hyperplasia. The latter two are currently believed to fall within the so-called “benign endometrial hyperplasia sequence” [103]. In a recent study, Aguilar and colleagues conducted genomic and IHC analysis of serial endometrial biopsies from patients with endometrial carcinoma, documenting the existence of endometrial lesions subdiagnostic of AH/EIN with marker aberrancy but that progressed to EIN or cancer [24]. IHC marker aberrancy was present in a significant proportion of these lesions—18% of disordered proliferative endometrium, 48% of non-atypical hyperplasia, and 50% of subdiagnostic lesions that defied precise classification. Genomic analysis further confirmed genetic alterations in those lesions with marker aberrancy. Importantly, cases with the marker aberrancy were more likely to progress to AH/EIN or endometrial carcinoma [78]. These findings underscore the need for further refinement of the current diagnostic criteria for AH/EIN, prompting important questions about whether the usage of marker panels should be extended to lesions beyond AH/EIN or those suspicious thereof. Should marker panels be applied to lesions as disordered proliferative endometrium and non-atypical hyperplasia, and should these lesions be subclassified based on the risk stratification via marker panels and managed differently? These questions necessitate further studies and consensus has yet to be reached.

## 9. Conclusions

The substantial progress in genomic and molecular investigations and biomarker studies has significantly enhanced our understanding of endometrial carcinoma and its precancers. This advancement is reflected in the evolution of diagnostic schemas over the past two decades. The incorporation of biomarkers into our daily clinical practice has been a gradual but steady process, leading to the updated diagnostic criteria in WHO 2020. Early investigations into the utility of the three-marker panel have compellingly shown its effectiveness in facilitating the accurate diagnosis of endometrial carcinoma precancers. Nevertheless, challenges persist and demand further exploration through future studies.

## Figures and Tables

**Figure 1 cancers-16-01159-f001:**
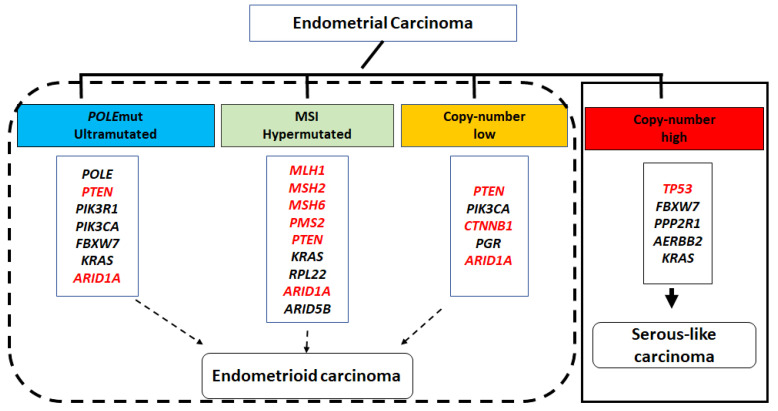
TCGA molecular classification and its approximate association with histopathological variants of endometrial carcinoma. The selected frequently mutated genes of each molecular subclass are listed underneath. The abnormal expression of genes highlighted in red has been investigated as a potential biomarker for atypical hyperplasia/endometrioid intraepithelial neoplasm (AH/EIN).

**Figure 2 cancers-16-01159-f002:**
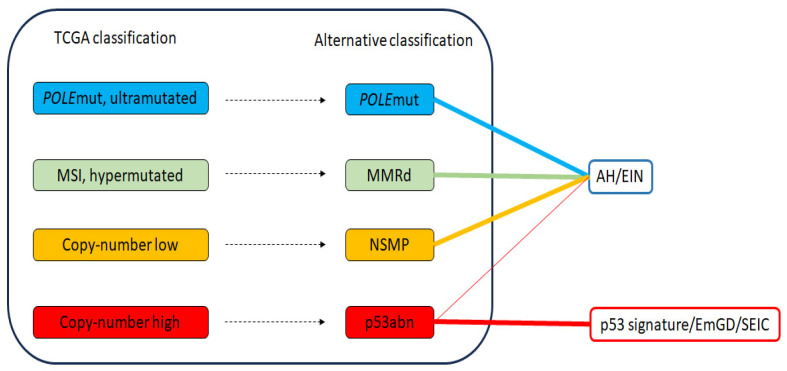
Relationship between TCGA molecular classification of endometrial carcinomas, alternative classification, and histopathologic variants of endometrial carcinoma precancers, along with the likely main pathway in the case of serous carcinoma. Dashed lines depict the approximate correlation between TCGA molecular classification and alternative classification groups of endometrial carcinomas. The thickness of the solid colored lines indicates the strength of association between endometrial carcinoma groups and their corresponding precancers; thicker lines represent stronger associations, while thinner lines indicate weaker associations. MSI—microsatellite instability; MMRd—mismatch repair deficient; NSMP—no specific molecular profile; abn—abnormal; mut—mutated; EmGD—endometrial glandular dysplasia; SEIC—serous endometrial intraepithelial carcinoma.

**Figure 3 cancers-16-01159-f003:**
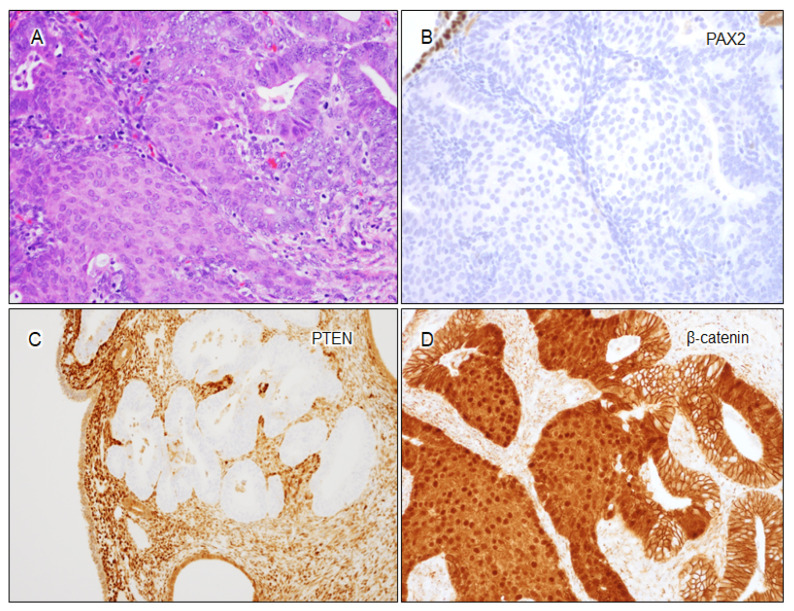
Illustration of morules and biomarker aberrancy in an AH/EIN case. (**A**) AH/EIN with morules (H&E, 200×); (**B**) loss of PAX2 expression (200×); (**C**) loss of PTEN expression (100×); (**D**) aberrant β-catenin nuclear staining (200×), with morules showing diffuse nuclear positivity for β-catenin.

**Figure 4 cancers-16-01159-f004:**
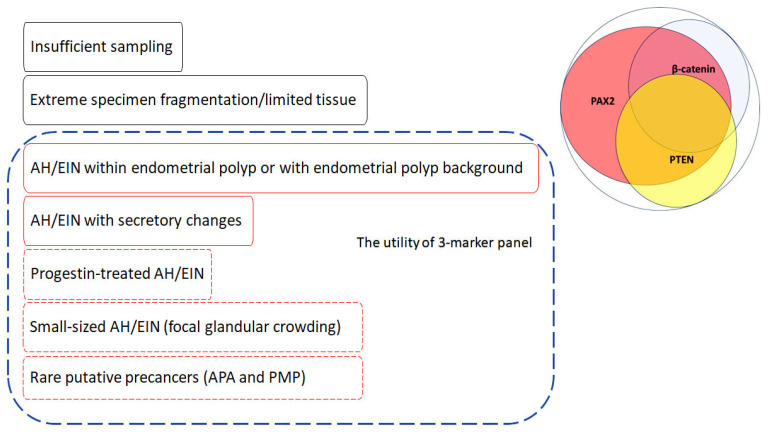
Practical challenges in achieving reliable diagnosis and the effectiveness of 3-marker-marker panel for atypical hyperplasia/endometrioid intraepithelial neoplasm (AH/EIN). Dashed blue frames enclose areas where the 3-marker panel has demonstrated either proven or potential utility. Areas with proven utility of the 3-marker panel are enclosed in solid red frames, while those with potential utility are enclosed in dashed red frames. Issues that cannot be resolved by biomarkers are enclosed in solid black frames.

**Figure 5 cancers-16-01159-f005:**
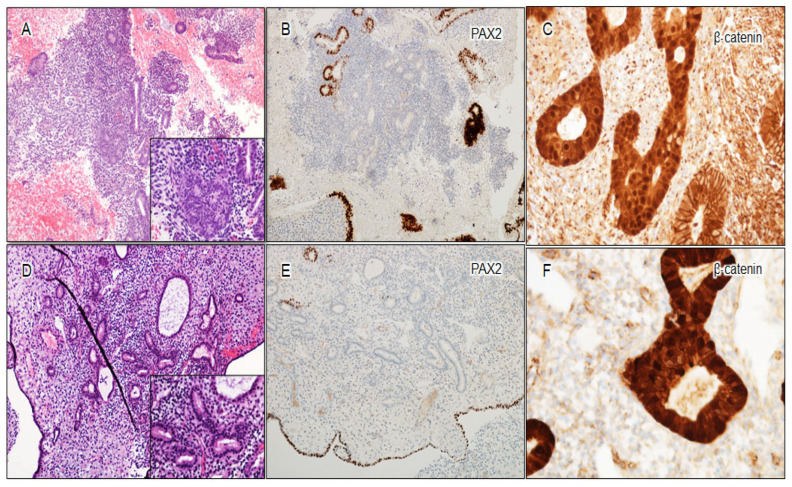
Illustration of biomarker aberrancy in small lesions. (**A**) Small-sized AH/EIN (H&E 40×, inset (200×)); (**B**) lesion exhibits the loss of PAX2 expression (40×); (**C**) lesion exhibits aberrant β-catenin nuclear staining (400×); (**D**) small residual AH/EIN (progestin-treated AH/EIN) (H&E 40×, inset (200×)); (**E**) residual AH/EIN exhibits the loss of PAX2 expression (40×); (**F**) lesion exhibits aberrant β-catenin nuclear staining (400×).

**Figure 6 cancers-16-01159-f006:**
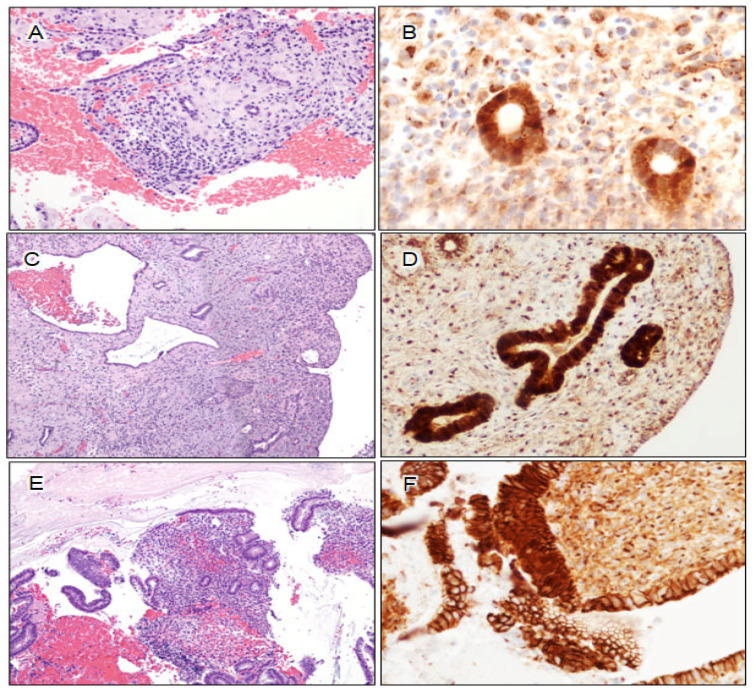
Examples of β-catenin aberrancy in seemingly “normal” endometrial glands during or after the cessation of progestin therapy. (**A**,**C**) H&E images of progestin-treated AH/EIN (40×). (**B**,**D**) β-catenin aberrancy in inactive-appearing endometrial glands in progestin-treated AH/EIN (400×). (**E**) H&E image of “normal” endometrium 6 months after the cessation of progestin treatment (40×). (**F**). β-catenin aberrancy in “normal” endometrium 6 months after the cessation of progestin treatment (400×).

**Figure 7 cancers-16-01159-f007:**
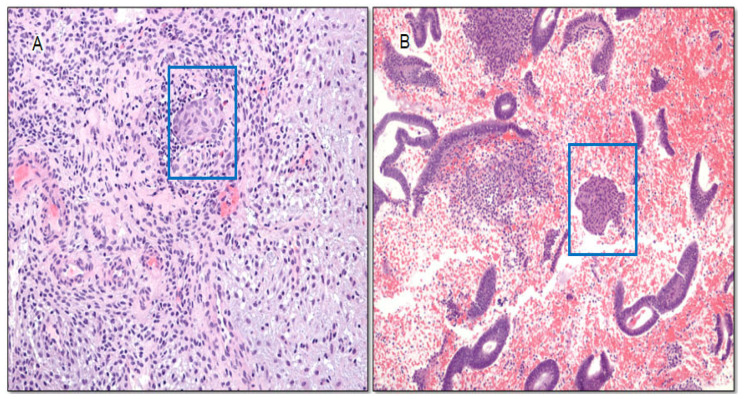
(**A**). Isolated morule in progestin-treated AH/EIN (H&E, 200×). (**B**) Isolated morule in endometrium 3 months after the cessation of progestin treatment (H&E, 200×). Blue boxes highlight morule.

**Table 1 cancers-16-01159-t001:** Expression characteristics of individual IHC markers in benign endometrium and AH/EIN.

	Staining Characteristics		
Markers	Normal Pattern	Aberrant Patterns in AH/EIN	Prevalence of Marker Aberrancy in AH/EIN
PAX2	Strong and uniform nuclear staining. Note: loss of nuclear staining in scattered glands is common in normal endometrium, typically in <5% of entire sample, except in cases of endometrial polyp, where approximately 10–20% of cases show significant PAX2 loss.	Large or diffuse lesion: absence of staining in > 5% of glands is considered as aberrant (strong and uniform expression in background normal glands served as internal control). Small lesion: >50% loss of expression in cytologically distinct glands of interest. In background of endometrial polyp: PAX2 loss should be interpreted with caution and in combination with morphology and other markers.	~70–80%
PTEN	Mainly cytoplasmic staining. Loss of expression in scattered glands is common in normal endometrium, typically in <5% of entire sample.	Large or diffuse lesion: the absence of staining in > 5% of glands was considered as aberrant (variable cytoplasmic expression in background normal glands served as internal control). Small lesion: >50% loss of expression in cytologically distinct glands of interest.	~40–50%
β-catenin	Ubiquitously membranous staining, except in some cases of interval phase endometrium, where weak nuclear localization can be seen.	Strong distinct nuclear expression (same density as cell membranes) may manifest as diffuse or focal. Within AH/EIN glands, nuclear staining can be uniform among all cells or only scattered cells are positive.	~50%
ARID1A	Ubiquitously nuclear staining.	Loss of nuclear staining.	~10%
Mismatch repair (MMR) proteins	Ubiquitously nuclear staining.	Loss of nuclear staining.	<5%
p53	Variable proportion of tumor cell nuclei staining with variable intensity.	Overexpression (involving at least 80% of tumor cell nuclei). Completely absent (null type). Cytoplasmic (unequivocal cytoplasmic staining accompanied by variable nuclear staining).	<3%

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
