# Peer review of "Biomarkers in the Diagnosis of Endometrial Precancers. Molecular Characteristics, Candidate Immunohistochemical Markers, and Promising Results of Three-Marker Panel: Current Status and Future Directions"

_cancers, 2024, doi:10.3390/cancers16061159_

Round 1

Reviewer 1 Report

Comments and Suggestions for Authors

The authors discussed, based on extensive literature (108 ref. including two articles on their own material) and in a comprehensive manner, the usefulness of a panel consisting of three IHC markers in the diagnosis of precancerous lesions of the uterine corpus. According to the current state of knowledge, a panel consisting of three markers (Pax2, PTEN, β-catenin) is an important complement to the histopathological diagnosis of AH/EIN.

I have no comments,  I read the article with interest.

Author Response

We appreciate greatly the generous evaluation by the reviewer. 

Reviewer 2 Report

Comments and Suggestions for Authors

This is a comprehensive review of the immunohistochemical panel for detecting atypical endometrial hyperplasia / EIN. This article describes the clinicopathological meaning of immunohistochemical markers representing molecular alteration. It provides useful information about the accurate diagnosis of AH/EIN. I enjoy reading the article; however, several points should be revised, as shown below.

1.        In page 11, figure legend for figure 4 is misaligned.

2.        In figure 4,5, pictures of β-catenin can be more improved.

3.        While authors emphasize PAX2, PTEN, and β-catenin, pictures of PTEN and ARID1A should be also presented.

4.        Line 417: “This finding suggests that APA and PMM” Does PMM mean PMP?

5.        If possible, a picture of β-catenin immunostaining in morules is desired in the article.

6.        While authors stress the importance of the 3-marker panel in the diagnosis of AH/EIN, detailed data for accurate diagnosis by number of marker aberrancies should be presented.

Author Response

This is a comprehensive review of the immunohistochemical panel for detecting atypical endometrial hyperplasia / EIN. This article describes the clinicopathological meaning of immunohistochemical markers representing molecular alteration. It provides useful information about the accurate diagnosis of AH/EIN. I enjoy reading the article; however, several points should be revised, as shown below.

  1. In page 11, figure legend for figure 4 is misaligned.

Response 1: Thank you for pointing this out. We realigned it accordingly.

  1. In figure 4,5, pictures of β-catenin can be more improved.

Response 2: Thank you for the suggestion. We replaced Figures 5 and 6 (previously Figures 4 and 5) with only high magnification images of β-catenin to highlight the nuclear staining pattern.

  1. While authors emphasize PAX2, PTEN, and β-catenin, pictures of PTEN and ARID1A should be also presented.

Response 3: Thank you for the suggestion. We have generated a new Figure 3 illustrating a case with all three-marker aberrancy and morules. Unfortunately, as ARID1A is not currently available in our clinical laboratory, we don’t have an ARID1A image to include this time.

  1. Line 417: “This finding suggests that APA and PMM” Does PMM mean PMP?

Response 4: Thank you for pointing this out. Indeed, it is PMP. We corrected it accordingly.

  1. If possible, a picture of β-catenin immunostaining in morules is desired in the article.

Response 5: Thank you for the suggestion. We generated a new Figure 3 illustrating a case with three-marker aberrancy and morules.

  1. While authors stress the importance of the 3-marker panel in the diagnosis of AH/EIN, detailed data for accurate diagnosis by number of marker aberrancies should be presented.

Response 6:  Thank you for the suggestion. 

In section 6 titled “Multi-marker panels for the identifying AH/EIN”, the following data are presented:

“A recent study by Aguilar et al. evaluated the efficacy of a 3-marker panel in identifying AH/EIN. The rate of individual marker positivity in AH/EIN varied with PAX2 exhibiting better sensitivity (81%) than PTEN (51%) or β-catenin (48%). Notably, the majority of cases (70%) exhibited aberrant expression in at least two markers, enhancing diagnostic reliability. When applied collectively, the combination of PAX2, PTEN, and β-catenin achieved a significantly improved sensitivity of 92.8% in bona fide AH/EIN detection.”

Reviewer 3 Report

Comments and Suggestions for Authors

This is an interesting study. However, some alterations should be made in the manuscript.

A. There are numerous mistakes in the text. 

10 Examples

Line 76: Correct Figure1 to Figure 1

Line 101: Correct mutations[11] to mutations [11]

Line 148:  independently[21,29] to  independently [21,29]

Line 153: Figure 2 (regular) to Figure 2 (bold)

Line 156: Candidate … to 5. Candidate

Line 212: 5.3β-. catenin to 5.3. β-catenin

Line 258: also known BAF250A to also known as BAF250A

Line 372: There is no caption for the Figure

Line 397: There is no Figure

Line 464: Unresolved to 8. Unresolved

B. In “copy-number high (serous-like)”, it should be emphasized that approximately 25% are endometrioid endometrial cancer G3 

Comments on the Quality of English Language

There are numerous mistakes in the text. 

10 Examples

Line 76: Correct Figure1 to Figure 1

Line 101: Correct mutations[11] to mutations [11]

Line 148:  independently[21,29] to  independently [21,29]

Line 153: Figure 2 (regular) to Figure 2 (bold)

Line 156: Candidate … to 5. Candidate

Line 212: 5.3β-. catenin to 5.3. β-catenin

Line 258: also known BAF250A to also known as BAF250A

Line 372: There is no caption for the Figure

Line 397: There is no Figure

Line 464: Unresolved to 8. Unresolved

Author Response

This is an interesting study. However, some alterations should be made in the manuscript.

- Response1 : We appreciate greatly the generous evaluation by the reviewer.

A. There are numerous mistakes in the text. 

10 Examples

Line 76: Correct Figure1 to Figure 1

Line 101: Correct mutations[11] to mutations [11]

Line 148:  independently[21,29] to  independently [21,29]

Line 153: Figure 2 (regular) to Figure 2 (bold)

Line 156: Candidate … to 5. Candidate

Line 212: 5.3β-. catenin to 5.3. β-catenin

Line 258: also known BAF250A to also known as BAF250A

Line 372: There is no caption for the Figure

Line 397: There is no Figure

Line 464: Unresolved to 8. Unresolved

-Response 2: Thank you for pointing those errors out. We made corrections accordingly. In addition, we went through the manuscript again to identify additional mistakes/formatting errors.

B. In “copy-number high (serous-like)”, it should be emphasized that approximately 25% are endometrioid endometrial cancer G3 

-Response 3: Thank you for the suggestion. A statement ” Approximately 25% of FIGO grade 3 endometrioid carcinomas are also classified as copy-number high tumors (molecular group 4).” was added to the modified manuscript.

There are numerous mistakes in the text. 

10 Examples

Line 76: Correct Figure1 to Figure 1

Line 101: Correct mutations[11] to mutations [11]

Line 148:  independently[21,29] to  independently [21,29]

Line 153: Figure 2 (regular) to Figure 2 (bold)

Line 156: Candidate … to 5. Candidate

Line 212: 5.3β-. catenin to 5.3. β-catenin

Line 258: also known BAF250A to also known as BAF250A

Line 372: There is no caption for the Figure

Line 397: There is no Figure

Line 464: Unresolved to 8. Unresolved

- Response 4: Thank you for pointing those errors out. We made corrections accordingly. In addition, we went through the manuscript again to identify additional mistakes/formatting errors.

Reviewer 4 Report

Comments and Suggestions for Authors

In the submitted review manuscript authors gave an overview on the present biomarkers in the diagnosis of endometrial precancers.

This manuscript is comprehensive, concise, and quite well written.

However, there are several mostly minor drawbacks which must be corrected or further improved:

1) Line 54: 2020 WHO Classification should be properly referenced.

2) Words "correlation" and "correlated" should be replaced with "association" and "associated", since they were not used in the context of a correlation coefficient.

3) Some text on Figure 1 seems blurry.

4) In 'Introduction', a brief explanation about what is considered as "precancer" and "early cancer" should be provided, since the text on that topic in lines 122-125, but also generally, is vague.

5) Figure 2 is vague and its legend is not exploratory enough, since it is unclear what those dashed and colored lines actually mean. Also, all abbreviations presented in figure should be explained in figure legend.

6) In lines 222 and 299 it is unclear if those numbers present references, since they were not put in square brackets.

7) Lines 211 and 232: β-catenin should be consistently written. Also, authors should re-check whole text that all gene symbols are always written and italics.

8) Sections were not properly numbered, since the main sections lack numbers.

9) The title of the figure should be removed from Figure 3, protein names on right part of the figure are not visible, and it should be explained what also those non-red frames present.

10) Ending sentence in lines 331-332 with "; however." seems awkward.

11) Figure 4 is not self-explanatory and its legend is not informative enough, i.e., it is unclear what this figure actually present. Also, it is unclear why IHC Figures 5 and 6 are so large while Figure 4 small. Furthermore, scale bar should be presented or at least magnification stated. However, since this is a review manuscript, it is unclear if those IHC figures are from other papers or were made particularly for this manuscript. If former, they must be properly referenced, and if latter, much more technical information must be provided for them.

Author Response

This manuscript is comprehensive, concise, and quite well written.

-We appreciate this comment.

However, there are several mostly minor drawbacks which must be corrected or further improved:

1) Line 54: 2020 WHO Classification should be properly referenced.

Response 1: Thank you for pointing this out. The reference has been added.

2) Words "correlation" and "correlated" should be replaced with "association" and "associated", since they were not used in the context of a correlation coefficient.

Response 2: Thank you for pointing this out. Corrected accordingly (highlighted in red)

3) Some text on Figure 1 seems blurry.

Response 3: Thank you for pointing this out. We increased the font size in the now Figure 1 to increase the visibility of the text.

4) In 'Introduction', a brief explanation about what is considered as "precancer" and "early cancer" should be provided, since the text on that topic in lines 122-125, but also generally, is vague.

Response 4: Thank you for the suggestion. The following sentence within Introduction was modified to clarify the definition of precancer. “The early and accurate detection of its precancers—precursor lesions with a high risk for progression to cancers—is of paramount importance, offering a valuable opportunity for a cure before progression to full-fledged cancer.”

5) Figure 2 is vague and its legend is not exploratory enough, since it is unclear what those dashed and colored lines actually mean. Also, all abbreviations presented in figure should be explained in figure legend.

Response 5: Thank you for the suggestion. A new figure legend was added with detailed explanation of Figure 2.

6) In lines 222 and 299 it is unclear if those numbers present references, since they were not put in square brackets.

Response 6: Thank you for pointing this out. Indeed, those numbers represent references. References have been appropriately added in the modified manuscript.

7) Lines 211 and 232: β-catenin should be consistently written. Also, authors should re-check whole text that all gene symbols are always written and italics.

Response 7: Thank you for the suggestion. We edited β-catenin and gene symbols throughout the manuscript.

8) Sections were not properly numbered, since the main sections lack numbers.

Response 8: Thank you for pointing this out. We corrected those errors accordingly.

9) The title of the figure should be removed from Figure 3, protein names on right part of the figure are not visible, and it should be explained what also those non-red frames present.

Response 9: Thank you for the suggestion. We modified the figure (now Figure 4) and legend accordingly.

10) Ending sentence in lines 331-332 with "; however." seems awkward.

Response 10: Thank you for this suggestion. “However” was deleted.

11) Figure 4 is not self-explanatory and its legend is not informative enough, i.e., it is unclear what this figure actually present. Also, it is unclear why IHC Figures 5 and 6 are so large while Figure 4 small. Furthermore, scale bar should be presented or at least magnification stated. However, since this is a review manuscript, it is unclear if those IHC figures are from other papers or were made particularly for this manuscript. If former, they must be properly referenced, and if latter, much more technical information must be provided for them.

Response 11: Thank you for pointing this out. It was a formatting error. We reformatted the manuscript so this figure comes with its legend.  We also modified the figure legends for Figures 4-6 (now Figure 5-7) to accommodate the suggested technical/formatting aspects.